# Deblurring of Sound Source Orientation Recognition Based on Deep Neural Network

**DOI:** 10.3390/s22207909

**Published:** 2022-10-18

**Authors:** Tong Wang, Haoran Ren, Xiruo Su, Liurong Tao, Zhaolin Zhu, Lingyun Ye, Weitao Lou

**Affiliations:** 1School of Earth Sciences, Zhejiang University, Hangzhou 301127, China; 2Explosion and Seismic Sensing Research Center, Advanced Technology Institute, Zhejiang University, Hangzhou 301127, China; 3Hainan Institute, Zhejiang University, Sanya 572025, China; 4College of Biomedical Engineering and Instrument Science, Zhejiang University, Hangzhou 301127, China

**Keywords:** deep neural network, beamforming, deblurring, sound source, orientation recognition

## Abstract

Underwater target detection and identification technology are currently two of the most important research directions in the information disciplines. Traditionally, underwater target detection technology has struggled to meet the needs of current engineering. However, due to the large manifold error of the underwater sonar array and the complexity of ensuring long-term signal stability, traditional high-resolution array signal processing methods are not ideal for practical underwater applications. In conventional beamforming methods, when the signal-to-noise ratio is lower than −43.05 dB, the general direction can only be vaguely identified in the general direction. To address the above challenges, this paper proposes a beamforming method based on a deep neural network. Through preprocessing, the space-time domain of the target sound signal is converted into two-dimensional data in the angle-time domain. Subsequently, we trained the network with enough sample datasets. Finally, high-resolution recognition and prediction of two-dimensional images are realized. The results of the test dataset in this paper demonstrate the effectiveness of the proposed method, with a minimum signal-to-noise ratio of −48 dB.

## 1. Introduction

With the development of precision detection instruments, the accurate detection and identification of short-range targets became possible [1,2]; therefore, scholars have turned to the positioning and identification of long-distance signals. Beamforming is an effective method to identify long-distance target signals, which has been widely used in radar, MIMO and other fields [3]. The general classical beamforming method is based on Fourier transform in the directional domain, where the input signal-to-noise ratio condition must be satisfied to ensure high resolution. The conventional beamforming method (CBF) achieves better adaptability by exploiting the phase deviation without prior knowledge of the number of signal sources, such as minimum variance distortion-free response (MVDR) and multiple signal classification (MUSIC) [4,5]. Although there are many beamforming methods for long-distance signals, the conditions of a high signal-to-noise ratio generally cannot be met. Furthermore, challenges remain in the detection of underwater targets under conditions of low signal-to-noise ratio, particularly when large distances exist between the sensor array and the target.

In recent years, with the rise of artificial intelligence and neural networks, an increasing number of fields have introduced and applied deep learning to practical problems and achieved satisfactory results. Deep learning was first applied to speech recognition and image segmentation and has since been further developed beyond its limitations. For example, Dong et al. proposed the neural network framework of the U-Net model [6], which integrates motion information into the network input and gives different motion constraints for each pixel; compared with the best-performing algorithm PSNR (peak signal-to-noise ratio), its value was improved by 0.14 dB. Hu et al. introduced a novel end-to-end U-Net and SAS network (U-SAS-Net) to extract and fuse local and semantic hierarchical features from a relatively large receptive field, resulting in an improved PSNR by 0.6 dB [7]. Zhang et al. used the improved U-Net deep neural network to extract data features through compression channels and restore data details through expansion channels, forming a nonlinear mapping from noisy data to denoised data, and finally outputting the denoised results [8]. Chen proposed an automatic segmentation algorithm for brain tumor MRI images based on an improved U-Net. Introducing residual blocks in U-Net to replace convolution blocks and adding attention modules to skip connections enables the model to converge faster during training and focus on the area to be segmented to obtain a better actual segmentation effect [9]. Wang proposed a superpixel segmentation method based on the U-Net network. This method embeds a norm layer after the convolution layer of each feature scale in the U-Net network, which is used to enhance the sensitivity of the network to parameters and effectively improve the segmentation accuracy of the medical image superpixels [10]. Jahn developed a method based on a neural network that can jointly estimate the spectral masks of all frequencies, and then estimate the cross-power spectral density matrix of speech and noise, so that the signal and noise can be accurately identified during beamforming [11]. Therefore, the neural network has an immeasurable ability for target recognition. Compared with the fuzzy recognition of conventional beamforming, if the deep neural network is combined with beamforming, it may obtain better results. Thus, this paper combines the U-Net network with beamforming, which can accurately identify the signal under the condition of a low signal-to-noise ratio. The lowest signal-to-noise ratio is improved from −40.6 dB to −48 dB compared to the traditional method. In addition, the deep neural network realizes the conversion of graph to graph, that is, both the input and the output are two-dimensional images. For the learning (training) process of a neural network, what the dataset learns is particularly important, as the quality of the dataset determines the learning of the network and the quality of the prediction results.

## 2. Method and Implement

### 2.1. Traditional Beamforming Methods

In the traditional methods, the CBF is used for angular decomposition, but this method has insufficient spatial resolution and poor noise adaptability. Therefore, in 2011, T.C. Yang introduced a deconvolution algorithm called Richardson–Lucy for high-power spectral estimation [12]. Deconvolution has been widely used in geological mapping and other complex environments [13]. Su et al. proposed a subspace vector deconvolution (SVD) method based on the deconvolution algorithm to improve its robustness [14].

In the frequency domain, depending on how the signal arrives, the receiver gets the signal from:(1)$$R=V*S_{\mathrm{source}}+N_{\mathrm{noise}}$$
where V represents a matrix of steering vectors with M rows and N columns; S_source_ denotes the source signal of length N; N_noise_ denotes the noise in the background and N_noise_ is the same size as V. In the time domain, N_noise_ is generally bigger than V ∗ S_source_. To eliminate the disadvantage brought by the N_noise_ matrix, a Fourier series is used to highlight the frequency characteristics in Equation (2):(2)$$X_{R}\left(k\right)=\sum_{n=0}^{N-1}R\left(n\right)\exp^{-j\left(2\pi /N\right)\mathrm{kn}}$$

The received data in Equation (1) can be expressed by the following Equation (3):(3)$$P=S_{\mathrm{ideal}}⊗K+N_{G}$$
where ⊗ means a Kronecker product; P replaces R as mentioned above, which stands for the blurred data; S_ideal_ represents the ideal signal in the signal process; K represents PSF with the same size as V; N_G_ is white Gaussian noise.

For Poisson noise distribution, the likelihood probability of the desired S_ideal_ can be expressed as:(4)$$p\left({P|S}_{\mathrm{ideal}}\right)P=\prod_{x}^{N}\left(S_{\mathrm{ideal}}⊗K\right)x^{B\left(x\right)}\exp[-\left(S_{\mathrm{ideal}}⊗K\right)x]B\left(x\right)!$$
(5)$$B_{P}\left(x\right)=\mathrm{Poisson}\left(\left(S_{\mathrm{ideal}}⊗K\right)\left(x\right)\right)$$

In order to obtain the solution of the above equation, the power function needs to be minimized:(6)$$S_{\mathrm{ideal}}{}^{*}=\mathrm{argmin}\;E\left(S_{\mathrm{ideal}}\right)$$
(7)$$E\left(S_{\mathrm{ideal}}\right)=\sum \left[\left(S_{\mathrm{ideal}}⊗K\right)-P\times \log(S_{\mathrm{ideal}}⊗K\right)]$$

Equation (7) is called Kullback–Leibler divergence (relative entropy). The Lucy–Richardson algorithm needs two prerequisites: received data, PSF, and ideal data must be nonnegative; the first two data must integrate to 1. The prerequisites mean that the RL algorithm has two features: nonnegative and energy retention. By controlling the preconditions, the iteration of the solution can be expressed as:(8)$$S_{\mathrm{ideal}}{}^{t+1}=S_{\mathrm{ideal}}{}^{t}\left[K^{P/\left(S_{\mathrm{ideal}}{}^{t}⊗K\right)}\right]$$
where t represents the number of iterations, which varies with different inputs. For the original discrete Fourier transform, a discrete signal of finite length can be expressed as:(9)$$X\left(k\right)=\sum_{n=0}^{N-1}x\left(n\right)Y_{N}^{\mathrm{kn}},k=0,\;1,\ldots ,N-1$$
(10)$$Y_{N}^{\mathrm{kn}}=e^{-j2\pi /N}$$

Then, the PSF can be replaced by:(11)$$X_{k}=\sum_{i=1}^{l}{\left|{\left(Y_{N}^{\mathrm{kn}}\right)}^{T}v_{j}Y_{N}^{\mathrm{kn}}\right|}^{2}$$
where l denotes the length of data.
(12)$$P_{k}=\sum_{i=1}^{M}X_{k}w^{H}A$$

The Plancherel theorem can acquire the strongest direction of all sensors in Equation (12), in which P_k_ represents the power of beamforming depending on different theta, w is the weight vector of the beamforming, and A corresponds to Equation (2).

### 2.2. Processing of Array Angle Recognition Based on Neural Network

U-Net was originally proposed by Ronneberger in 2015 and applied to medical cell segmentation [15]. Prior to that, the field of computer vision used fully connected (FCN) for image segmentation. Ronneberger proposed a more ‘elegant’ fully convolutional network, whose structure is divided into two parts, an encoder and a decoder, and the shape is exactly like the ‘U’ letter, thus it was named U-Net.

The following part briefly introduces the two important components of U-Net, namely, the encoder and the decoder.

Assuming that the initial input image size is a 572 × 572 grayscale image, after two convolution kernels of 3 × 3 × 64 (64 3 × 3 convolution kernels, to obtain 64 feature maps), the convolution operation becomes 568 × 568 × 64 in size. Then, a 2 × 2 max pooling operation is performed to yield 248 × 248 × 64, and the above process is repeated four times. Each time the first 3 × 3 convolution operation after pooling is performed, the number of 3 × 3 convolution kernels exponentially increases. When the bottom layer is reached, that is, after the 4th maximum pooling, the image becomes 32 × 32 × 512 in size, and then two 3 × 3 × 1024 convolution operations are performed, finally resulting in a 28 × 28 × 1024 size. The above operations constitute the downsampling process.

At this time, the size of the image is 28 × 28 × 1024. Firstly, we perform a 2 × 2 deconvolution operation to change the image to a 56 × 56 × 512 size. Then, we copy and crop the image before the corresponding maximum pooling layer, and deconvolve the image obtained by stitching it together to get a 56 × 56 × 1024 size image, and then perform a 3 × 3 × 512 convolution operation. We repeat the above process four times, and in the first 3 × 3 convolution operation after each stitching, the number of 3 × 3 convolution kernels is doubled. When the top layer is reached, that is, after the 4th deconvolution, the image becomes 392 × 392 × 64 in size. The image is copied and cropped, and then stitched to obtain a size of 392 × 392 × 128. Subsequently, two 3 × 3 × 64 convolution operations are performed to get an image of 388 × 388 × 64 size, and finally, we perform a 1 × 1 × 2 convolution operation. The above steps constitute the upsampling process.

The deep neural network used in this paper has a similar structure to the classic U-NET (Figure 1) and is also divided into four downsamplings and four upsamplings. Each convolution kernel is 3 × 3 × 64, the pooling layer is 2 × 2, and the activation function is ReLU.

Initially, the sampling frequency was set to 500 Hz, the underwater sound speed was 1500 m/s, the total length of the array was 1 km, the number of array elements was 101 equally spaced in the 1 km array, and the sampling time was 3600 s.

Ship trajectory generation process: within random distances of 20–100 km, and a random angle of −55~+55, distance-time domain datasets were generated. These are then preprocessed by the conventional beamforming (CBF) method and converted into the azimuth-time domain. The azimuth(angle) and time are the horizontal and vertical axes of the image, respectively. The time sampling rate is set to every 30 s, and the angle sampling rate is set to every 1 degree, that is, a 121 × 120 image is generated, and this image is used as the original input of the network. The actual trajectory of the ship is used as a network label (known). Then we put the original input datasets and network labels into the training module, where the total number of samples is 100 (1000 samples and 100 samples are pre-generated for testing; the network learning effect is the same, because the sample generation takes a long time (one sample takes nearly 300 s), so using 100 samples can save a lot of time and still achieve the training effect). The training set is 70%, the validation set is 30%, and the epoch is set to 200.

## 3. Data Example

### 3.1. Noise-Free Conditions

The preprocessed time-angle map is used as the input of the network, and the known real ship trajectory map (that is, the real time-angle map) is taken as the label of the network and put into network training.

It can be seen from the Figure 2 that the CBF method and the deconvolution method have low resolution (the recognition result is shadowy and blurry) for the identification of the ship’s trajectory (azimuth), and they do not meet the current requirements for high-precision detection. Meanwhile, the U-Net neural network used in this paper greatly improves the angular resolution through its training, and the final prediction results are basically consistent with the original real data (clear and without shadows), which proves the effectiveness of the network.

### 3.2. Noisy Conditions

Considering that in practical applications, the array reception is affected by a series of environmental factors, it is not practical to test only at a noise-free level, therefore, we take the actual noise into account.

Marine environmental noise is caused by natural and human factors, such as sea surface wind and waves, underwater undercurrent surging, ship operation, wind power project construction and operation periods, and submarine construction operations. The sound wave signal forms a relatively complex background noise field after reflection and absorption associated with seawater, sea level, seabed, etc. The academic community generally believes that ship noise and windborne noise are the two main components of this noise. The former mainly affects the noise in the middle and low frequency bands (10–500 Hz), and the latter mostly influences the noise in the higher frequency bands (500 to 25,000 Hz). In addition to the above two major components, in the low frequency band (1–100 Hz), tides, surges, waves, large-scale turbulence, and distant earthquakes and storms all contribute to marine environmental noise [16,17,18].

Finally, referring to the article published by Wenz [14] in 1962, we regard wind noise as environmental noise, which is simulated according to the empirical formula summed up by the Wenz curve for the wind noise of different energy levels. The empirical formula of wind noise is as follows:(13)G(f)=−A∗lg(f)+b

Among them, G(f) represents the final noise level in dB; f represents the frequency; A and b are affected by the environment, and the values of A and b are different in different environments. For the convenience of the experiment, we take different values of A and b respectively, which represent the noise of different energy levels.

The above empirical formula is the wind noise on the sea surface. In fact, the sensors we deploy are generally under the sea surface, where the impact of the wind noise is not significant, so the noise added to the simulation is larger.

Finally, different levels of noise are added, and the signal-to-noise ratio is introduced (the same energy level of noise, the distance between the sensor position, and the distance of propagation will eventually lead to a large difference in the signal-to-noise ratio), and the advantages and disadvantages of the traditional method and the network method are compared.

#### 3.2.1. Test One—When the Signal-to-Noise Ratio Is −26.78 dB (Relatively High)

When the value of A in the empirical formula is 14.2 and the value of b is 86.3, the noise level is relatively small, and the signal-to-noise ratio is relatively high.

It can be seen from the Figure 3 that when the noise with the b value of 86.3 in the empirical formula is added, the SNR is −26.78 at this time. The deconvolution method and the input of the network (data after CBF) can roughly identify the angle, but the angle resolution is not high, and the prediction result of the neural network is completely consistent with the label (real data), showing that the network is very helpful for the improvement of angular resolution, and the prediction accuracy is also high.

#### 3.2.2. Test Two—When the Signal-to-Noise Ratio Is −33.57 dB~−37.18 dB (Relatively Slightly Lower)

When the value of A in the empirical formula is 14.2, and the value of b is 91.3, it is equivalent to increasing the noise and reducing the signal-to-noise ratio.

It can be seen from the Figure 4 that when the noise with the b value of 91.3 energy level in the empirical formula is added, the deconvolution method and the input of the network can also identify the approximate angle, and both show a gradual blurring trend; however, the network can still improve the angle resolution, that is, the network output is still highly accurate, and the prediction accuracy is also extremely high.

#### 3.2.3. Test Three—When the Signal-to-Noise Ratio Is −40.6 dB~−43.05 dB (Relatively Low)

Here, the value of A in the empirical formula is 14.2 and the value of b is 96.3.

It can be seen from the Figure 5 that when the noise with the b value of 96.3 in the empirical formula is added, the information identified by the deconvolution method is overwhelmed by the noise, and the trajectory cannot be identified at all by the naked eye. At this time, the network learning ability is not greatly affected.

#### 3.2.4. Test Four—When the Signal-to-Noise Ratio Is −44.27 dB~−50.35 dB (Relatively Extremely Low)

Here, the value of A in the empirical formula is 14.4, and the value of b is 96.3.

It can be seen from the Figure 6 that when the signal-to-noise ratio is lower than −43 dB, the network’s ability to recognize the angle gradually decreases, and there are artifacts. Until the signal-to-noise ratio reaches −50 dB, the network’s recognition ability is completely invalid.

After sorting out the experimental data, the concept of accuracy is introduced, that is, we correlate the prediction results of CBF, deconvolution, and U-NET with the real trajectory (label), and if the error range between the result and the real trajectory is within the spatial resolution (1 degree), the prediction/recognition is considered accurate, and conversely, the prediction/recognition is wrong. Finally, the ratio of all correctly predicted quantities to total quantities is calculated. Figure 7 below shows the accuracy change of different methods under different signal-to-noise ratio conditions.

To sum up, when the values of A and b in the empirical formula are different, the signal-to-noise ratio changes greatly, and the difference between the deconvolution method and the network identification method is also large. Among them, under the condition of no noise, both the traditional method and the proposed network can learn the label data well, and the prediction result is consistent with the label. Meanwhile, after adding noise, when the signal-to-noise ratio is higher than −33.5 dB, the identification of the network is consistent with the label, and the traditional deconvolution method is also effective; when the signal-to-noise ratio is lower than −33.5 dB, the deconvolution method gradually blurs and fails, but the network identification is still accurate, until the signal-to-noise ratio is as low as −43 dB. The information in the deconvolution method and the U-Net network input is completely overwhelmed by noise; the deconvolution method basically fails, and at this time, the accuracy of the network decreases slightly, and some small artifacts appear in the network identification. When the signal-to-noise ratio is lower than −43 dB, the network identification ability gradually declines, but it can still identify a part of the trajectory information; until the noise ratio is lower than −50 dB, the network fails completely.

## 4. Discussions and Conclusions

The orientation recognition of underwater sound sources is a hot research topic in hydroacoustics. The existing conventional beamforming method (CBF) has a certain efficiency in the identification of ship trajectories, but there are still problems, such as inaccuracy and low resolution. In this paper, we use the U-Net deep neural network to train and learn the preprocessed time-angle ship trajectories and put the pregenerated label dataset into the network. According to the final prediction results of the network, in the absence of noise, the U-Net network can accurately identify the specific orientation of the ship’s trajectory. Compared with CBF, the angular resolution is greatly improved. After adding wind noise, when the signal-to-noise ratio is higher than −43 dB, the U-Net network can learn and predict the true trajectory of the ship; when the signal-to-noise ratio is lower than −43 dB, the noise is completely drowned out by the true trajectory, and the U-Net network can also partially identify the true trajectory of the ship, accompanied by a few artifacts. When the signal-to-noise ratio reaches −50 dB, the learning ability of the network becomes completely invalid.

The deconvolution methods compared in this paper have certain conditions of applicability conditions, that is, when the difference between the energy level of the target and the energy level of the interference is large (usually 40 dB), these methods are effective, but for the background noise simulated in this paper, they are no longer applicable. At the same time, the neural network method proposed in this paper is more accurate and effective. Therefore, compared with the traditional method, our U-Net neural network method reduces the lower limit of the signal-to-noise ratio to −48 dB, which has a significant effect on the improvement of angular resolution.

Compared with the traditional methods, the neural network is data driven and does not rely on a priori assumptions, but has requirements on a known label dataset. The deep neural network used in this paper is based on U-Net. Subsequent experiments may verify the recognition of the target trajectory angle under noisy conditions by different types of neural networks, and compare the similarities, differences, advantages, and disadvantages of each network.

## Figures and Tables

**Figure 1 sensors-22-07909-f001:**
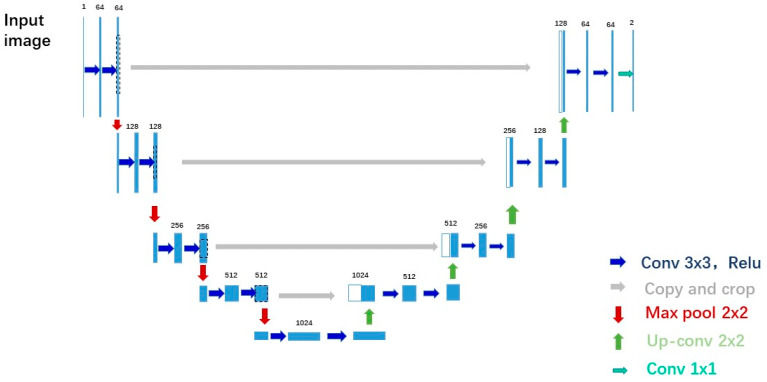
U-Net Network Structure Diagram.

**Figure 2 sensors-22-07909-f002:**
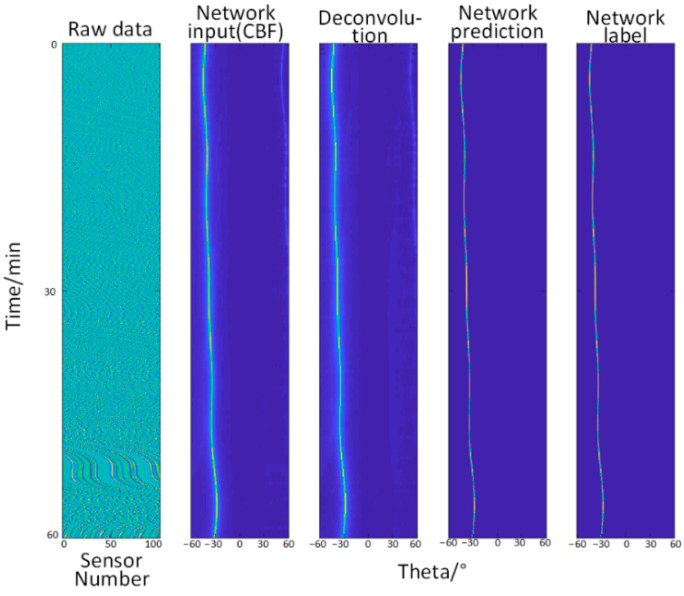
Comparison of angle recognition results by different methods under noise-free conditions, the input of the network (the abscissa is the angle, and the ordinate is the time, which are the same below), deconvolution method, the prediction of the network, and the network label (true trajectory).

**Figure 3 sensors-22-07909-f003:**
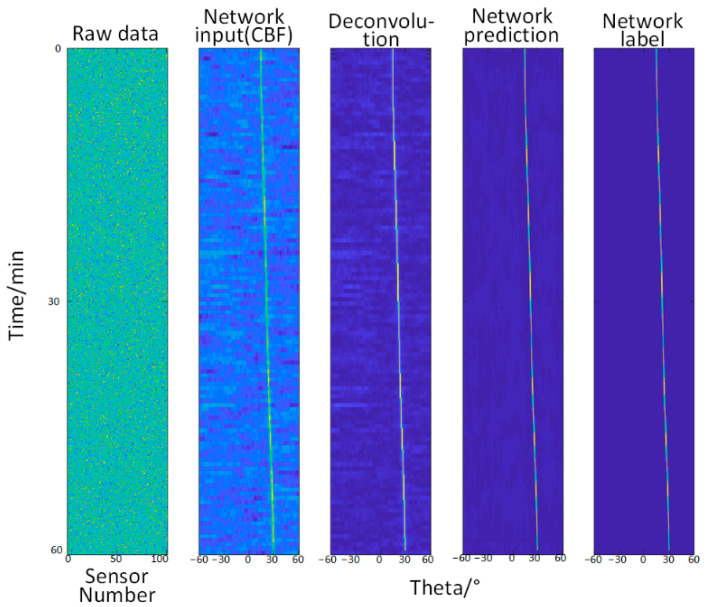
Comparison of angle recognition results between different methods when the signal-to-noise ratio is −26.78. From left to right: raw data (the abscissa is the sensor number, and the ordinate is the time), the input of the network (the abscissa is the angle, and the ordinate is the time, the same below), deconvolution method, the prediction of the network, and the network label (true trajectory).

**Figure 4 sensors-22-07909-f004:**
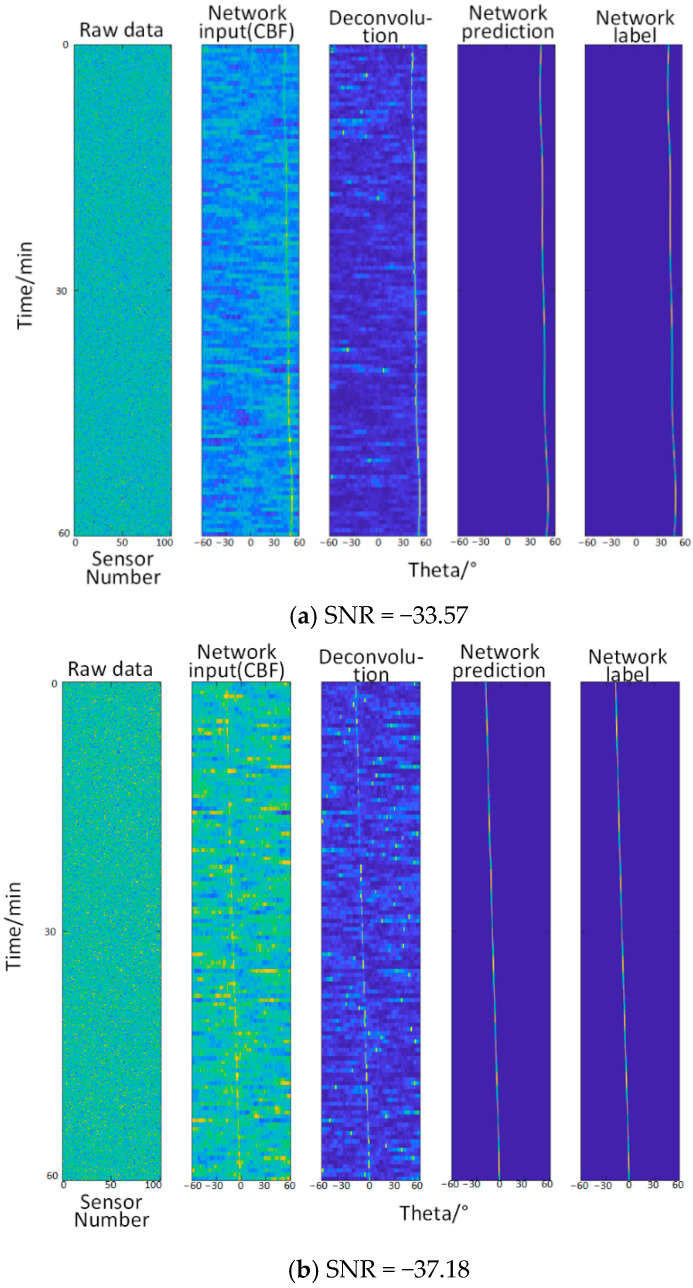
(**a**,**b**) are respectively the angle recognition results and prediction results of different methods under different signal-to-noise ratio conditions. (**a**,**b**) from left to right: original data (the abscissa is the sensor number, and the ordinate is the time), the input of the network (the abscissa is the angle, and the ordinate is the time, which is the same below), deconvolution method, the prediction of the network, and the network label (true trajectory).

**Figure 5 sensors-22-07909-f005:**
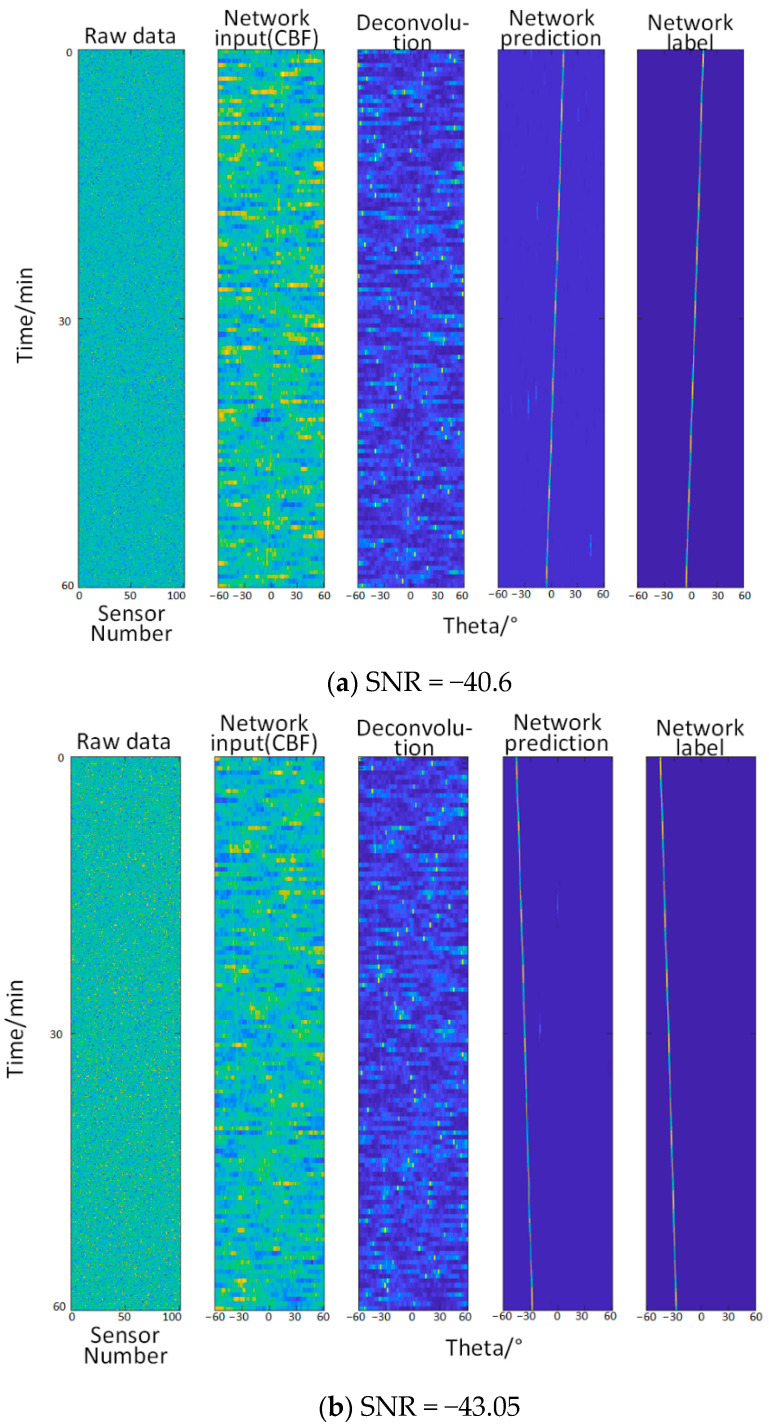
Comparison of angle recognition results of different methods for different levels of signal-to-noise ratio. From left to right: original data (the abscissa is the sensor number, and the ordinate is the time), the input of the network (the abscissa is the angle, and the ordinate is the time, which is the same below), deconvolution method, network prediction, network label (true trajectory).

**Figure 6 sensors-22-07909-f006:**
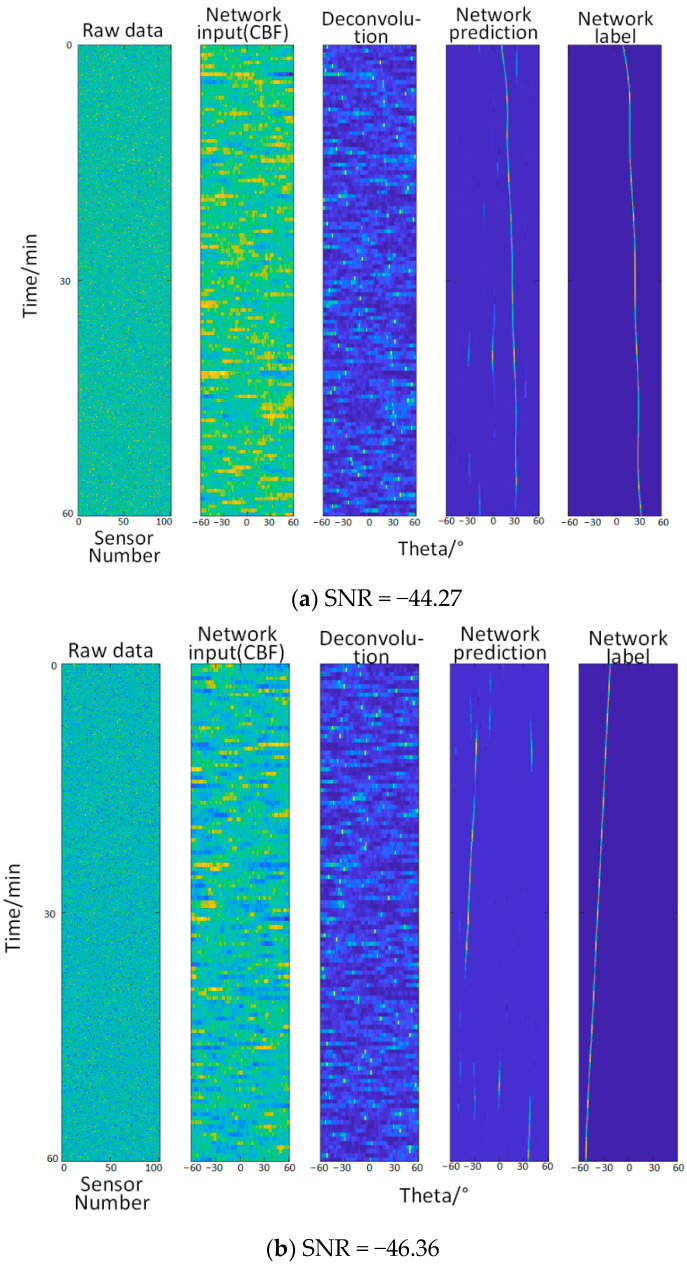
(**a**–**d**) show the angle recognition results and prediction results of training different methods under different signal-to-noise ratio conditions. (**a**–**d**) from left to right: original data (the abscissa is the sensor number, and the ordinate is the time), the input of the network (the abscissa is the angle, and the ordinate is the time, which is the same below), deconvolution method, the network predictions, network labels (true trajectories).

**Figure 7 sensors-22-07909-f007:**
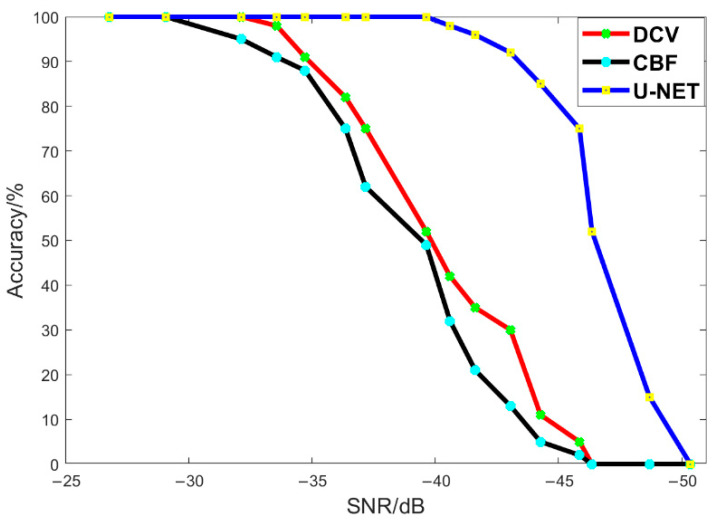
Graph of accuracy versus signal-to-noise ratio, in which the red line represents the deconvolution method, the black line represents the CBF method, and the blue line represents the U-Net method proposed in this paper.

## Data Availability

The data that support the findings of this study are available from the corresponding author upon reasonable request.

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
