# Peer review of "Deblurring of Sound Source Orientation Recognition Based on Deep Neural Network"

_sensors, 2022, doi:10.3390/s22207909_

Round 1

Reviewer 1 Report

your article is quite good and have a novelty in your field research, i just need some adjustment in your research:

1. Give equation/formula/function from figure 2-6 and show process from figure

2. need more references/literature or make some of previous research.

Author Response

Dear Reviewer:

   Thank you for your valuable comments on this article, what follows is my explanation of your question

  1. Thank you for your valuable comments. Figure 2-6 contains original data, deconvolution method, CBF, original input, network label, network output; the derivation process of deconvolution method has been presented in the paper, if not detailed, please refer to this article: DOI: https:/doi.org/10.3390/s22062327. Of course, its author is also one of the authors of this article. I will upload the program of the U-NET method used in this article to the editor if necessary.
  2. The list of references has been expanded.

These are all my answers, if you have any questions please contact me again, thank you again for your patience.

Sincerely

Tong Wang

[email protected]

Reviewer 2 Report

In this paper, deep learning is applied to the problem of estimating the direction of arrival of underwater sound to obtain performance beyond that of conventional methods. This is an interesting study for system development. However, the academic novelty is somewhat low, and the explanation of the evaluation results is somewhat insufficient.

Please review the comments listed below.

1. It is difficult to understand the intent of the explanation of the beamforming method in 2.1. Is this used to create the U-NET input data for this study?

2. In relation to comment 1, the explanation in 2.1 was difficult for me to understand. I would like to see a more detailed explanation of the formulas. In particular, the definitions of variables and operators need to be better defined.

3. This question is beyond the scope of this paper, but why did you choose the beamforming results computed with CBF as the network input in this study instead of the original signal?

I think deep learning, which converts acoustic echoes into time and angle information, would be more effective in practice.

4. I don't think you have explained how you changed the SNR for Figures 4, 5 and 6 (forgive me if I missed something).

5. It was not clear from what perspective or intent the parameters (A, b) for each test were set. It is difficult to interpret the evaluation results obtained if we do not have some understanding of the specific environment in which these conditions were set.

6. Figure 7 is not well explained. I think that ``Accuracy'' needs to be defined.

7. Why not try to solve this problem with new ideas, not just U-NET? The point that a true trajectory is needed should be improved. I think that just one twist to this would increase the novelty of this study.

Author Response

Dear Reviewer:

      Thank you for your valuable comments on this article, what follows is my explanation of your question

  1. Thank you for your valuable comments.Sorry for not explaining it clearly before. The beamforming and deconvolution methods compared in this article are described in Section 2.1. The U-NET input data in this article is a two-dimensional image processed by CBF.
  2. Thank you for your valuable comments. The explanation of deconvolution in Section 2.1 is not clear enough, please refer to this article: DOI: https:/doi.org/10.3390/s22062327. Of course, its author is also one of the authors of this article.
  3. For this problem, the purpose of this paper is to optimize the processing results of CBF after the basis of CBF processing, that is, deblurring; it is not directly using the network to process the original data.
  4. Regarding changing the SNR of Figures 4, 5, and 6, it has been explained in the original text: According to the empirical formula obtained from the Wenz curve of wind noise at different energy levels, it can be simplified to G(f)=-A*lg(f)+ b. During the experiment, the added wind noise energy level is adjusted by modifying the values of A and b in the formula, and the final program will also output the SNR value.
  5. Point 4 has already been explained.
  6. Thank you for your valuable comments. The accuracy rate is not strictly defined in the article. The meaning of the accuracy rate refers to the ratio of the part of the real trajectory that the network output can identify to the network label (100%, no noise). For example, the network can restore half of the trajectory, which is 50%.
  7. Thank you for your valuable comments.Because this article aims to use the U-NET network to process the image deblurring after CBF, it is a graph-to-graph conversion. Since U-NET has been very successful in the training and learning of 2D images, it is selected. But in subsequent articles, I will experiment with neural network processing results in different environments such as different types of noise, multi-source signals, etc., and of course, I will compare and use other neural networks.

The above is my answer. Thank you again for your patient review and correction. If there are still unclear points, please communicate with me again.

Sincerely

Tong Wang

[email protected]

Round 2

Reviewer 2 Report

I personally feel that the novelty of this study is somewhat lacking.

However, I understand that you have proposed a method with sufficient performance for practical use.

Therefore, I judge this manuscript to be accepted.

Author Response

Dear Reviewer:

       Thank you very much for taking the time to review my article twice and give very valuable advice.

Regarding the novelty of the paper, the method in question has achieved application innovation in the field of hydroacoustics. I believe that subsequent research based on this method will be more helpful in the field of hydroacoustics.

Regarding the method description part of the paper you mentioned, I have made targeted changes in accordance with the editor's comments.

Sincerely

Tong Wang

[email protected]